# Resonant Forcing of the Climate System in Subharmonic Modes

**Jean-Louis Pinault** 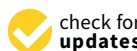

Independent Scholar, 96, rue du Port David, 45370 Dry, France; jeanlouis_pinault@hotmail.fr;
Tel.: +33-7-89-94-65-42

**Abstract:** During recent decades observation of climate archives has raised several questions. Concerning the mid-Pleistocene transition problem, conflicting sets of hypotheses highlight either the role of ice sheets or atmospheric carbon dioxide in causing the increase in duration and severity of ice age cycles. The role of the solar irradiance modulations in climate variability is frequently referenced but the underlying physical justifications remain most mysterious. Here, we extend the key mechanisms involving the oceanic Rossby waves in climate variability, to very long-period, multi-frequency Rossby waves winding around the subtropical gyres. Our study demonstrates that the climate system responds resonantly to solar and orbital forcing in eleven subharmonic modes. We advocate new hypotheses on the evolution of the past climate, implicating the deviation between forcing periods and natural periods according to the subharmonic modes, and the polar ice caps while challenging the role of the thermohaline circulation.

**Keywords:** resonantly forced baroclinic waves; subharmonic modes; climate variability

## 1. Introduction

The paleoclimate records show perfect coherence between temperature transitions and orbital forcing but with effects not proportional to the presumed causes [1–6]. On the other hand, variations in the Earth's radiative budget cannot, as such, explain climatic variability, whatever the time scale. As already suggested [7] an amplifying and filtering effect remains to be found to explain the resonant nature of the global climate system. Still unexplained climatic oscillatory features, which are ubiquitous and can be observed regardless of the frequency band [8], are addressed with a new approach that generalizes the key mechanisms involving the baroclinic waves in climate variability [9–14] to long-period, multi-frequency 'Gyral Rossby Waves' (GRWs).

Nondispersive Rossby waves of very long-period (tens of thousands years) are hypothesized by lengthening while winding around the subtropical gyres, proportionally to the period, the apparent wavelength of 8-yr average period Rossby waves dragged by the wind-driven inertial recirculation. These are observed wherever the western boundary currents leave the continents to re-enter the interior flow of subtropical gyres [8]. The concept of GRWs is supported by both: (1) the observation of long-period Sea Surface Temperature (SST) anomalies around the North Atlantic subtropical gyre (2) the solution of the equations of motion obtained by using the β-cone approximation so that the gradient β of the Coriolis parameter depends on the mean radius of the gyre, and remains constant all around the latter. The speed of the anticyclonically wind-driven inertial recirculation being higher than the phase velocity of cyclonically propagating GRWs, as a result of solar forcing resonances occur when half the apparent wavelength of GRWs is an integer number of times the perimeter of the gyre.

A positive feedback occurs in a loop on the velocity of the modulated polar current around the gyre and the amplitude of the thermocline oscillation because of the temperature difference between low and high latitudes. GRWs have a direct impact on climate as a consequence of the

acceleration/deceleration of the western boundary current. The duration of heating/cooling during half a period of the upper ocean around the gyre compensates for wave damping due to Rayleigh friction as the period increases, both being proportional to the period with antagonistic effects. Applying the Caldirola–Kanai equation of coupled oscillators with inertia to long-period, multi-frequency GRWs sharing the same modulated polar current around the gyre, subharmonic resonances occur [15]. This approach being more constraining than that of delayed oscillators [16,17], the solution specifies the subharmonic modes of oscillators, namely the number of turns of the apparent half-wavelength allowed around the gyres. The solar forcing efficiency of GRWs depends straightforwardly on the deviation between their natural periods according to the subharmonic modes, and the forcing period.

In this study, we will elucidate: (1) the reason why the forcing efficiency varies according to the three orbital parameters; (2) the conditions in which the dominant subharmonic mode has changed during the Pleistocene; (3) how the forcing efficiency varies during the Holocene; (4) how sudden cooling occurred at the beginning of the interglacial, reflecting the dominant subharmonic mode.

## 2. Method

### 2.1. GRWs and Surface Temperature

The methodological approach of this study supposes that the global temperature of the Earth deduced from the climate archives in both hemispheres is proportional to the mean amplitude of the corresponding GRWs, which is justified in the following way. Considering the unperturbed state of the global climate system for which the average energy captured by the Earth is completely re-emitted in space, the perturbed state resulting from the imbalance in energy budget due to heat transfer to high-latitudes of the gyres via the western boundary currents behaves like a quasi-isolated thermodynamic system [8]. This is because thermal exchanges are mainly ruled by latent and sensible heat fluxes (the contribution of radiative fluxes is low). This supposes that the perturbed state is restricted to the exchanges ocean-atmosphere-continents at mid-latitudes. So the heat energy accumulated in the gyre is conserved on a planetary scale. Compared to the unperturbed system, the perturbed state of the system tends to a new steady state in which a new thermal equilibrium occurs between the oceans and the continents. Then, the persistence of the oceanic perturbation $\Delta T$ reflects the renewal of the mixed layer at high latitudes of the gyres while maintaining the vertical temperature gradient. Within the perturbed state of the climate system it follows that the SST anomalies at high latitudes of the gyres tend to balance with the surface temperature perturbation of the continents (the surface temperature in the perturbed system averaged over a few years compared to that in the unperturbed system). This is because the thermal exchanges that occur within the perturbed state are fast in comparison with the processes that tend to blur the temperature perturbation of the continents to restore the temperature of the unperturbed state, while being sustained. This contrast between the kinetics of heat exchanges generates a dynamic equilibrium that tends to mitigate the spatial disparities. Therefore, everything happens as if the perturbed state of the global climate system was deduced from the unperturbed state so that the resulting perturbation of the global surface temperature is a replica of the perturbation $\Delta T$ of SST anomalies at high latitudes of gyres.

The impact on climate of the oceanic perturbation $\Delta T$, which either reinforce or, on the contrary, reduce evaporation depending on the sign of the perturbation $\Delta T$ of SST, results from atmospheric baroclinic instabilities that may lead to the formation of cyclonic or, on the contrary, anticyclonic systems. Baroclinic instabilities of the atmosphere are most active when the resulting synoptic-scale systems of high or low pressure are stimulated and guided by the subtropical jet streams in their respective hemisphere. Consequently, those SST anomalies are located at high latitudes of the gyres around 40°. Then, global processes take over to warm or cool the continents globally as a result of the long-period of the perturbation $\Delta T$.

In the quasi-isolated perturbed climate system the amplitude of the long-period Rossby waves can be considered as proportional to the variation of the surface temperature. This is how weighted sums

of SST anomalies averaged over relevant areas of subtropical gyres allow us to explain the natural component of the current climate variations, that is the regionalized instrumental surface temperature of the Earth before the anthropogenic component became prominent [18].

*2.2. Subharmonic Modes*

Sharing the same modulated polar current at the node where the western boundary currents leave the continents to re-enter the interior flow of the subtropical gyres, multi-frequency Rossby waves are coupled. Indeed if the polar velocity $u_i$ of a free Rossby wave is lower than the resultant velocity of the flow where the nodes of all Rossby waves are merging, the polar current $u_i$ of the coupled Rossby wave at the common node is faster than that of the free Rossby wave. The opposite happens when the polar velocity $u_i$ of a free Rossby wave is higher than the resultant velocity of the flow at the common node. This results in an acceleration/deceleration of the geostrophic currents associated with the coupled Rossby waves.

Resonant forcing always outweighs the non-resonant forcing that does not allow capturing as much energy. As occurs in the general case of coupled oscillators with inertia resonantly forced, the latter are subject to a subharmonic mode locking giving the dynamic system an optimal stability [15,19]. Generation of climate oscillations through nonlinear subharmonic resonance in delayed oscillators has been studied, which supposes either feedback strength is modulated periodically or periodic external forcing is applied [16,17]. Although it applies to Rossby waves, this concept is too general and produces results that are not sufficiently constrained as regards the periods of the various subharmonics. It is more relevant to solve a system of equations of motion reflecting the dynamics of coupled oscillators with inertia. As in the case of delayed oscillators, prototype equations have a periodic external forcing term common to the $N$ coupled equations of motion corresponding to the $N$ coupled oscillators. But the solution shows that a subharmonic mode locking occurs and specifies how the periods of resonance are deduced by recurrence from one oscillator to another.

Subharmonic modes of coupled Rossby waves are found by solving the equation of the Caldirola–Kanai (CK) oscillator, which is a fundamental model of dissipative systems applied to damped harmonic oscillators [19]. In the present case, the equation of the CK oscillator is formulated to express the mode of coupling between several resonantly forced Rossby waves that share the same modulated polar current at the node.

Then, the set of equations of motion takes the following form:

$$\mathcal{M}_i \ddot{u}_i + \gamma \mathcal{M}_i \dot{u}_i + \sum_{j=1}^{N} J_{ij}\left(u_i - u_j\right) \; = \; I_i \cos(\Omega t) \tag{1}$$

where the restoring force just depends on the difference in velocities $u_i$ of the modulated polar currents associated with the oscillators: it vanishes when the polar currents of the free oscillators have the same speed. The damping parameter $\gamma$ of Rossby waves is referring to the Rayleigh friction, the inertia parameter $\mathcal{M}_i$ to the mass of water displaced during a cycle resulting from the quasi-geostrophic motion of the $i$-th oscillator and $J_{ij}$ measures the coupling strength between the oscillators $i$ and $j$. The right-hand side describes the periodic driving on the $i$-th oscillator with amplitude $I_i$ and the common frequency $\Omega$.

The solution $u_i$ related to the $i$-th oscillator (proportional to the amplitude of the thermocline oscillation as well as the velocity of the radial current) is such that its natural period $\tau_i$ is an integer number of years which is deduced by recurrence: $\tau_i \; = \; n_i \tau_{i-1}$ with $\tau_0 \; = \; T$ ($T = 1$ year). The observation of surface temperature variations from ice cores shows that $n_i \; = \; 2$ most of the time. The only exception is $768 \; = \; 3 \times 256 \; = \; 3 \times 2^8$ probably due to constraints resulting from the adjustment to orbital forcing [8].

The reference period $T$ and the first subharmonics result from the natural periods of baroclinic waves in the tropical oceans and the resulting equatorial geostrophic currents that merge with the western boundary currents. These tropical baroclinic waves form a single dynamic system, that is, a

quasi-stationary wave (QSW) with equatorial and off-equatorial antinodes where the thermocline oscillates up and down, and equatorial and off-equatorial nodes where the resulting modulated geostrophic currents are accelerated [20,21]. The QSW period is tuned to the forcing period by the off-equatorial waves which play the same role as the tuning slide of a wind instrument. In the Atlantic Ocean, the driving force of the annual QSW mainly results from easterlies during its westward phase propagation. In the Indian Ocean the 1/2-yr period QSW is resonantly forced by the biannual monsoon. Sharing the same node, namely the modulated component of the Equatorial Counter-Current, it is coupled with the 1-, 2- and 4-year period QSWs resulting from the 2nd, 3rd and 4th baroclinic mode Rossby and Kelvin waves. These coupled QSWs produce the Indian Oscillation Dipole (IOD). In the Pacific Ocean the quadrennial QSW whose period is subject to a large variability is coupled with the annual first baroclinic, fourth meridional mode Rossby wave, the latter being resonantly forced by easterlies. The driving force of the quadrennial QSW partly results from its coupling with the annual QSW that share the South Equatorial Current at the node. It is partly self-sustained by generating an El Niño event [21,22] at the end of its eastward phase propagation: evaporative processes make the thermocline rise while the equatorial Rossby wave is receding toward the west. An Eastern or a Central Pacific El Niño event occurs according to whether the quadrennial and the annual QSWs are in phase or not.

At mid-latitudes, the formation of Rossby waves relies on the adjustments of the stratified oceans to the periodic change in warm water mass carried by the western boundary currents from the tropics, as is shown by solving the equations of motion with relevant boundary conditions [8]. As a consequence of multi-frequency QSWs in the tropical oceans, which result in particular from higher-order baroclinic modes, western boundary currents have the potential ability to convey a succession of warm water masses at particular frequencies, thus producing the deepening or the rising of the thermocline where they leave the continents to re-enter the interior flow of the subtropical gyres. Indeed, short-period QSWs at mid-latitudes exhibit western and eastern antinodes as well as western and eastern nodes both in opposite phases: the western antinode follows the gyre whereas the eastern antinode is outside the gyre, escaping poleward. On the other hand, the phases of the western antinode and node are opposite. In such conditions, the western antinode takes the place of the eastern antinode after half a cycle, while a new western antinode is formed, in opposite phase versus the previous one. This leads to the transfer poleward of successive warm water masses.

The propagation mode of long-period GRWs is similar: the antinode around the gyre takes the place of the antinode outside the gyre after half a cycle, while a new antinode around the gyre is formed, in opposite phase versus the previous one that escapes poleward. But forcing of long-period GRWs now results from variations in solar irradiance. Also their different phases of propagation must be expressed in relation to the latter. As shown by the solution of the equations of motion [8], a quarter period before the maximum of the solar irradiance the modulated polar current is anticyclonic and the thermocline reaches its highest level. Half a period later, the modulated polar current is cyclonic and the thermocline reaches its lowest level. In this case, the radial current controls the oscillation of the thermocline. The solution is represented for the 128-year period GRWs, namely the perturbation $\eta$ of the surface height, the modulated polar current $u$ and the modulated radial current $v$, respectively [23].

To each apparent half-wavelength $\lambda_i/2$ associated with the $i$-th oscillator, the number of turns $N_i = \lambda_i/2L$ around the gyre ($L$ is the length of the median streamline of the gyre) is the order of the subharmonic mode. Extrapolation of the apparent half-wavelength of Rossby waves deduced from the wavelet power of SSH anomalies whose mean period is 8 years shows that, for the $128 = 2^7$ year period, $N_i = 2$ in the North and South Atlantic and in the North Pacific, $N_i = 1$ in the South Pacific and $N_i = 3/2$ in the south Indian Ocean [8].

## 2.3. Positive Feedback Loop

The efficiency of solar and orbital forcing results from the imbalance between incoming and outgoing fluxes through the surface of the ocean [8]. In the absence of resonance of long-period Rossby waves the inflow is balanced mainly by evaporation. By contrast, lowering of the thermocline and

acceleration of the convergent radial current while the intensity of forcing is increasing makes the wave behave as a heat sink because of downwelling. On the other hand, uprising of the thermocline and acceleration of the divergent radial flow while forcing intensity is decreasing makes the wave restores the accumulated heat: it then acts as a heat source due to upwelling.

The linear equations of motion do not involve the positive feedback that occurs in a loop on the polar current velocity and the thermocline depth as a result of the temperature difference between low and high latitudes of the gyre. This feedback occurs on the one hand when, a quarter period before the maximum solar irradiance, the polar current is anticyclonic. The heat flux transported poleward by the western boundary current favors the deepening of the thermocline, therefore the acceleration of the polar current whose speed reaches its maximum nearly at the same time as the maximum solar irradiance. On the other hand, a quarter period after the maximum solar irradiance, the polar current is cyclonic and the rise of the thermocline is favored by the low temperatures at high latitudes of the gyre. Uprising of the thermocline promotes the deceleration of the polar current and less heat is transported poleward by the western boundary current.

This positive feedback loop considerably increases the efficiency of solar and orbital forcing modulations resulting from the acceleration or deceleration of the modulated polar current of the gyre, that is to say the western boundary current, while anticipating the oscillation of the thermocline by nearly a quarter period. Furthermore, for a given radiative power modulation the amplitude of the thermocline oscillation increases with the temperature gradient between low and high latitudes of the gyre, which reinforces the positive feedback. As a corollary, the western boundary current is strongly controlled by the extension of the polar ice caps. This deduction does not involve the thermohaline circulation as a driver, but it behaves here like an overflow.

## 3. Data

Proxies of the global temperature are obtained from foraminifera in sediment cores [24] and from the analysis of ice cores. Deuterium data $^2$H obtained from Antarctica Dome C ice core (European Project for Ice Coring in Antarctica EPICA) are used for global mean temperature estimate in the southern hemisphere [25]. $^{18}$O data obtained from Greenland Summit Ice Cores GISP2 (Greenland Ice Sheet Project 2 Ice Core) [26,27] are used as proxies of global mean temperature in the northern Atlantic.

In ice cores, the proportions of different oxygen and hydrogen isotopes provide information about ancient temperatures. The isotopic composition of the oxygen in a core can be used to model the temperature history of the ice sheet. The ratio between $^{18}O$ and $^{16}O$ indicates the temperature when the snow fell. Hydrogen ratios can also be used to calculate a temperature history. The deuterium excess reflects the temperature, relative humidity, and wind speed of the ocean where the moisture originated.

Calcareous fossil Foraminifera are formed from elements found in the ancient seas where they lived. They can be used, as a climate proxy, to reconstruct past climate by examining the stable isotope ratios and trace element content of the shells. Global temperature can be revealed by the isotopes of oxygen.

$^{10}$Be and $^{14}$C which are stored in polar ice cores and tree rings, offer the opportunity to reconstruct the history of solar activity. In the series obtained by Steinhilber et al. [28] different $^{10}$Be ice core records from Greenland and Antarctica are combined with the global $^{14}$C tree ring record using principal component analysis to provide total solar irradiance variations (W/m$^2$).

The production rates of $^{14}$C and $^{10}$Be vary in the same way because the two isotopes are produced by similar processes in the atmosphere. However, the $^{14}$C concentration also is influenced by changes in the global carbon cycle. Therefore the $^{10}$Be-$^{14}$C synchronization involves a modelling step to account for the influence from the carbon cycle on the $^{14}$C concentration.

Data on changes in the Earth's orbital parameters and the resulting variations in insolation were calculated by Berger [29].

## 4. Wavelet Analysis

By decomposing a time series into time–frequency space, wavelet analysis allows determination of both the dominant modes of variability and how those modes vary in time [30]. In the present context, the wavelet power spectrum allows representation of the variation of insolation over time. Furthermore the cross-wavelet power spectrum of both variation of insolation and Earth's temperature allows us to represent the joint variation of the two variables over time. To examine fluctuations in power over a range of scales (periods), the scale-averaged wavelet power is defined as the weighted sum of the wavelet power spectrum over a band. The software is available at http://paos.colorado.edu/research/wavelets/software.html.

## 5. Results

From the climate archives cited in "Data", the methodological approach aims at representing the variations in the global surface temperature as well as in the solar forcing within the contiguous bands of resonance of GRWs corresponding to each subharmonic mode. The forcing occurs when both variables are coherent with a small dephasing. The natural periods of GRWs have the property of tuning finely to the forcing period by a latitudinal shifting of their centroid, in accordance with the dispersion relation of free waves [8]. According to (6) in [8] the phase velocity $\omega/k$ of GRWs ($\omega$ is the pulsation, $k$ the wave number) is expressed as a function of the latitude $\varphi_0$ of the centroid of the gyre. The phase velocity decreases as the centroid shifts poleward.

This supposes a latitudinal elongation of the gyre to maintain the conditions of resonance of higher subharmonic modes. Since the bandwidth is substantial, much wider than the natural broadening of the forcing or the surface temperature cycles, any coherence between the two series during several contiguous cycles cannot be fortuitous. The comparison of the oscillations between them then helps affirm or deny that there is a causal relationship and appraise the forcing efficiency over time in the case a coupling is proven. Otherwise, the GRW is coupled to higher subharmonic modes.

As displayed in Table 1 eleven subharmonic modes are to be investigated to characterize the behavior of GRWs according to their natural periods, extending from 64 years to 98.3 Kyr. This table specifies the successive frequency bands studied, representative of the subharmonic modes and summarizes the main results that are to be discussed. GRWs do not show resonance beyond 98.3 Kyr. It appears that the $3 \times 2^{11}$ subharmonic mode whose period is 393.2 Kyr is not coupled with the eccentricity variations of 360 Kyr period, although it is more intense than that of 100 Kyr period. The only plausible reason seems to be the deviation between the natural and forcing periods, more than 30 Kyr at present. Tuning of the natural period of the GRWs to the forcing period would require a $1°15'$ equatorward drift of their centroid, which would likely jeopardize resonances of lower subharmonic modes.

**Table 1.** The natural periods of Gyral Rossby Waves (GRWs), and the bands characteristic of subharmonic modes. Subharmonic modes in the North and South Pacific have to be divided by 2 and multiplied by 3/2 in the South Indian Ocean.

| Band Width (yr) | Period of Resonance (yr) | Subharmonic Mode in the Atlantic | Forcing Mode |
|---|---|---|---|
| 48–96 | 64 | $2^0$ | No external forcing [18] |
| 96–192 | 128 | $2^1$ | Solar forcing (Gleissberg cycle) |
| 192–576 | 256 | $2^2$ | No external forcing |
| 576–1152 | 768 | $3 \times 2^2$ | Solar forcing |
| 1152–2304 | 1536 | $3 \times 2^3$ | No external forcing |
| 2304–4608 | 3072 | $3 \times 2^4$ | No external forcing |
| 4608–9216 | 6144 | $3 \times 2^5$ | No external forcing |
| 9216–18,432 | 12,288 | $3 \times 2^6$ | No external forcing |
| 18,432–36,864 | 24,576 | $3 \times 2^7$ | Orbital forcing (Precession) |
| 36,864–73,728 | 49,152 | $3 \times 2^8$ | Orbital forcing (Obliquity) |
| 73,728–14,7456 | 98,304 | $3 \times 2^9$ | Orbital forcing (Eccentricity) |

### 5.1. The Glacial-Interglacial Period

Each band associated with the last three subharmonic modes, namely $3 \times 2^7$, $3 \times 2^8$ and $3 \times 2^9$, corresponds to orbital forcing, the precession, the obliquity and the eccentricity, whose periods are centered on 23.5, 41 and 100 Kyr, respectively. The resonant nature of the climate system appears in two different ways. Firstly, the efficiency of orbital forcing depends heavily on the deviation between the forcing and the natural periods of GRWs so that the amplitude of GRWs do not reflect that of forcing. Secondly, the sharpness of the tuning between the natural and the forcing periods for the eccentricity—the deviation is less than 2%—makes the forcing efficiency depends heavily on the deviation between the two periods so that a competition occurs between the orbital forcing resulting from the obliquity and from the eccentricity, despite the large deviation between their amplitudes.

Because of the inherent limitations of the windowed Fourier transform the wavelet power spectrum of the series is represented, using the Morlet wavelet while being time-averaged over the whole interval of observation [30] (a) each component is normalized (b) real spectrum. (c) Cross-wavelet power spectrum of orbital variation of insolation (OVI) integrated over 65° S to 65° N [29] and EPICA [25] using the Morlet wavelet. The color gamut is applied to the covariance values included between 0 and 95% of the total covariance, that is, the covariance time-averaged, scale-integrated over the whole interval of frequencies, namely 1.65 °C × W/m². The black contour encloses regions of more than 95% confidence level. (d) Wavelet power spectrum of OVI. The variance time-averaged, scale-integrated over the whole interval of frequencies is 3.08 $W^2/m^{-4}$. The arrows point to the period of the eccentricity. (e–h) Coupling of the eccentricity variations with the global temperature within the band 73.7–147.5 Kyr—(e) unfiltered and filtered global temperature obtained from foraminifera in sediment cores (signals are centered) [24]. Wavelet-filtered time series are obtained by scale-averaging over the bandwidth, the wavelet transform—(f) unfiltered and filtered OVI—(g) comparison of filtered signals of Effective Radiative Forcing (ERF) and global temperature—(h) forcing efficiency that, multiplied by the forcing, gives the ERF. (i–l) Same as (e–h) applied to ice cores (EPICA). (m,n) Coupling of the obliquity variations with the global temperature within the band 36.9–73.7 Kyr. (o,p) Coupling of the precession variations with the global temperature within the band 18.4–36.9 Kyr. (q,r) Centered global temperature obtained from EPICA in (q) and filtered in (r).

As shown in the Fourier power spectrum of orbital parameters (Figure 1a,b), the forcing periods exhibit narrow bandwidths. The strongest orbital forcing results from the obliquity, the weakest from the precession. The cross-wavelet power of the orbital variations of insolation (OVI) and the global mean temperature estimated from EPICA shows that the coupling mainly occurs into three bands centered on 23 Kyr (precession), 41 Kyr (obliquity) and 100 Kyr (eccentricity), the first two being not resolved: Figure 1c.

#### 5.1.1. The 73.7–147.5 Kyr Band

The proxy of the global temperature is obtained from sediment cores in Figure 1e–h and from ice cores in Figure 1i–l. The forcing efficiency varies significantly during the period of observation (Figure 1h,l), gradually increasing from 0.7 to 5.0 °C × (W/m²)⁻¹ from 1400 Kyr BP to present. The coupling relaxes between 450 and 350 Kyr BP when the forcing collapses while the oscillation of the GRW is maintained, which suggests the remanence of large-scale geostrophic forces (Figure 1g,j). The increase in efficiency of forcing reflects that the period of orbital forcing and the natural period of GRWs become closer over time: the forcing period approaches 100 Kyr during the last million years (Figure 1d). Only a 15′ poleward drift of the latitude of the centroid of the GRW enables the tuning of the natural and forcing periods, which suggests the forcing efficiency is close to its optimum at present.

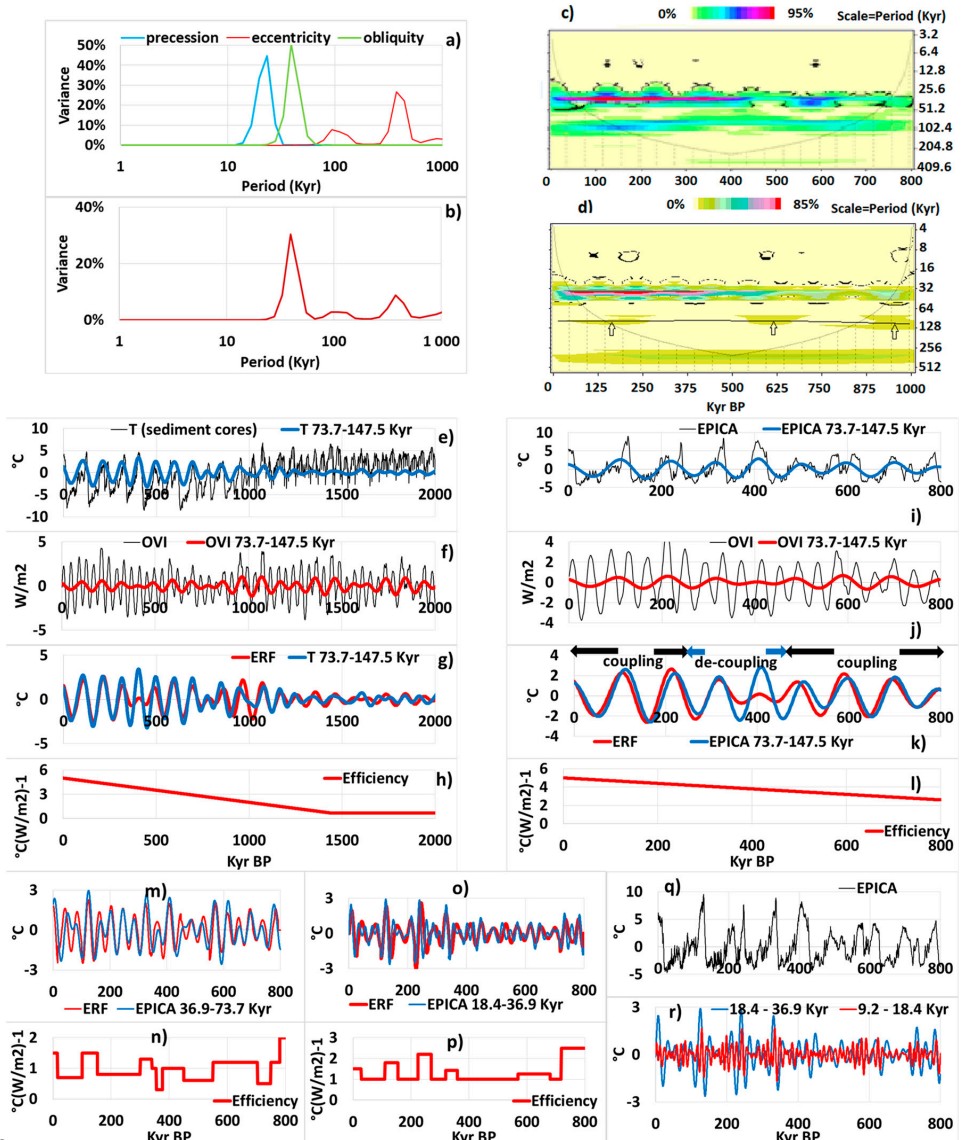

**Figure 1.** (**a,b**) Fourier power spectrum of the orbital parameters [29]. Because of the inherent limitations of the windowed Fourier transform the wavelet power spectrum of the series is represented, using the Morlet wavelet while being time-averaged over the whole interval of observation [30] (**a**) each component is normalized. (**b**) real spectrum. (**c**) Cross-wavelet power spectrum of orbital variation of insolation (OVI) integrated over 65°S to 65°N [29] and EPICA [25] using the Morlet wavelet. The color gamut is applied to the covariance values included between 0 and 95 % of the total covariance, that is, the covariance time-averaged, scale-integrated over the whole interval of frequencies, namely 1.65 °C×W/ m2. The black contour encloses regions of more than 95% confidence level. (**d**) Wavelet power spectrum of OVI. The variance time-averaged, scale-integrated over the whole interval of frequencies is 3.08 W2/m-4. The arrows point to the period of the eccentricity. (**e–h**) Coupling of the eccentricity variations with the global temperature within the band 73.7-147.5 Kyr (**e**) unfiltered and filtered global temperature obtained from foraminifera in sediment cores (signals are centered) [24]. Wavelet-filtered time series are obtained by scale-averaging over the bandwidth, the wavelet transform (**f**) unfiltered and filtered OVI (**g**) comparison of filtered signals of Effective Radiative Forcing (ERF) and global temperature (**h**) forcing efficiency that, multiplied by the forcing, gives the ERF. (**i–l**) Same as (**e–h**) applied to ice cores (EPICA). (**m,n**) Coupling of the obliquity variations with the global temperature within the band 36.9–73.7 Kyr. (**o,p**) Coupling of the precession variations with the global temperature within the band 18.4–36.9 Kyr. (**q,r**) Centered global temperature obtained from EPICA in (**q**) and filtered in (**r**).

### 5.1.2. The 36.9–73.7 Kyr Band

The forcing efficiency related to the obliquity undergoes abrupt transitions (Figure 1m,n). Lower than 1 °C × (W/m²)⁻¹ during periods of lull, the forcing efficiency exceeds 1.0 up to 1.5 °C × (W/m²)⁻¹ during times of culmination like this has happened several times, between 100 and 150 Kyr BP in particular. Consequently, the fine tuning of the natural period of the GRW to the forcing period occurs only when the amplitude of the latter enables the centroid to drift 2°30′ to the equator. This reflects that the GRW is subject to orbital forcing while being coupled to the $3 \times 2^9$ subharmonic mode.

### 5.1.3. The 18.4–36.9 Kyr Band

The orbital variations related to the precession, although low, force the GRW (Figure 1o,p) because of the proximity of the natural and forcing periods. Here again, the forcing efficiency is subject to strong variations in amplitude, varying between 1 and 2.5 °C × (W/m²)⁻¹. The fine tuning of the natural period of the GRW to the forcing period occurs during times of culmination while the amplitude of the forcing enables the centroid to drift 40′ to the equator. Outside these periods, the forcing efficiency remains around 1 °C × (W/m²)⁻¹ as occurs between 370 and 720 Kyr BP. As in the previous band, the GRW is subject to orbital forcing while being coupled to the $3 \times 2^9$ subharmonic mode.

### 5.1.4. The 2.3–4.6, 4.6–9.2 and 9.2–18.4 Kyr Bands

These bands are characterized by the weakness of the orbital forcing (Figure 1b). As shown in Figure 1q,r, the amplitude of the GRW in the 9.2–18.4 Kyr band varies considerably, being correlated with the $3 \times 2^7$ subharmonic mode. This phenomenon is especially noticeable since 350 Kyr BP, which suggests that the GRW is a pure harmonic without significant external forcing.

### 5.2. The Holocene

During the last interglacial, the modulated polar currents associated with the $3 \times 2^9$ and $3 \times 2^8$ subharmonic modes are anticyclonic because the global temperature modulation is positive (Figure 1k,m) while vanishing for the $3 \times 2^7$ subharmonic mode (Figure 1o). Consequently, the western boundary currents are supported by large-scale geostrophic forces. These powerful currents that modulate the wind-driven inertial recirculation are themselves modulated by the geostrophic polar currents resulting from lower subharmonic modes.

During the Holocene, the resonances of GRWs result from the variations in the Total Solar Irradiance (TSI) that reflect cycles related to the internal dynamics of the Sun. Unlike orbital forcing the Fourier power spectrum of the TSI exhibits a broad spectrum, with a peak centered on 935 years (Figure 2a). Consequently, the tuning of the natural period of the GRWs to the forcing period is much less sharp here and it is more difficult to differentiate the drivers, namely the solar forcing and the coupling with higher subharmonic modes. The evolution of the forcing efficiency depends heavily on the temperature gradient of the upper ocean between the low and high latitudes of the gyres, which is conditioned by the latitude of the ice cap boundary.

### 5.2.1. The 576–1152 yr Band

Figure 2b shows a coupling occurs between the TSI and the global temperature into the 576–1152 yr band of the $3 \times 2^2$ subharmonic mode, mainly within the interval covering 9000 to 6000 years BP. Although of lower amplitude, the solar forcing also occurs within the 96–192 yr band of the $2^1$ subharmonic mode, more discontinuously.

The observations from the North Atlantic in Figure 2k–m are essentially similar to those from Antarctica (Figure 2h,j). In the South hemisphere, the forcing efficiency (Figure 2j) is equal to 3 °C × (W/m²)⁻¹ between 9000 and 6500 years BP, and then decreases approximately to 1 °C × (W/m²)⁻¹ in the North Atlantic. At the beginning of the Holocene, the pack ice extends further south, which explains

the high forcing efficiency. Then, gradual withdrawal of the pack ice makes the positive feedback that occurs in a loop on the polar current velocity and the thermocline depth less efficient.

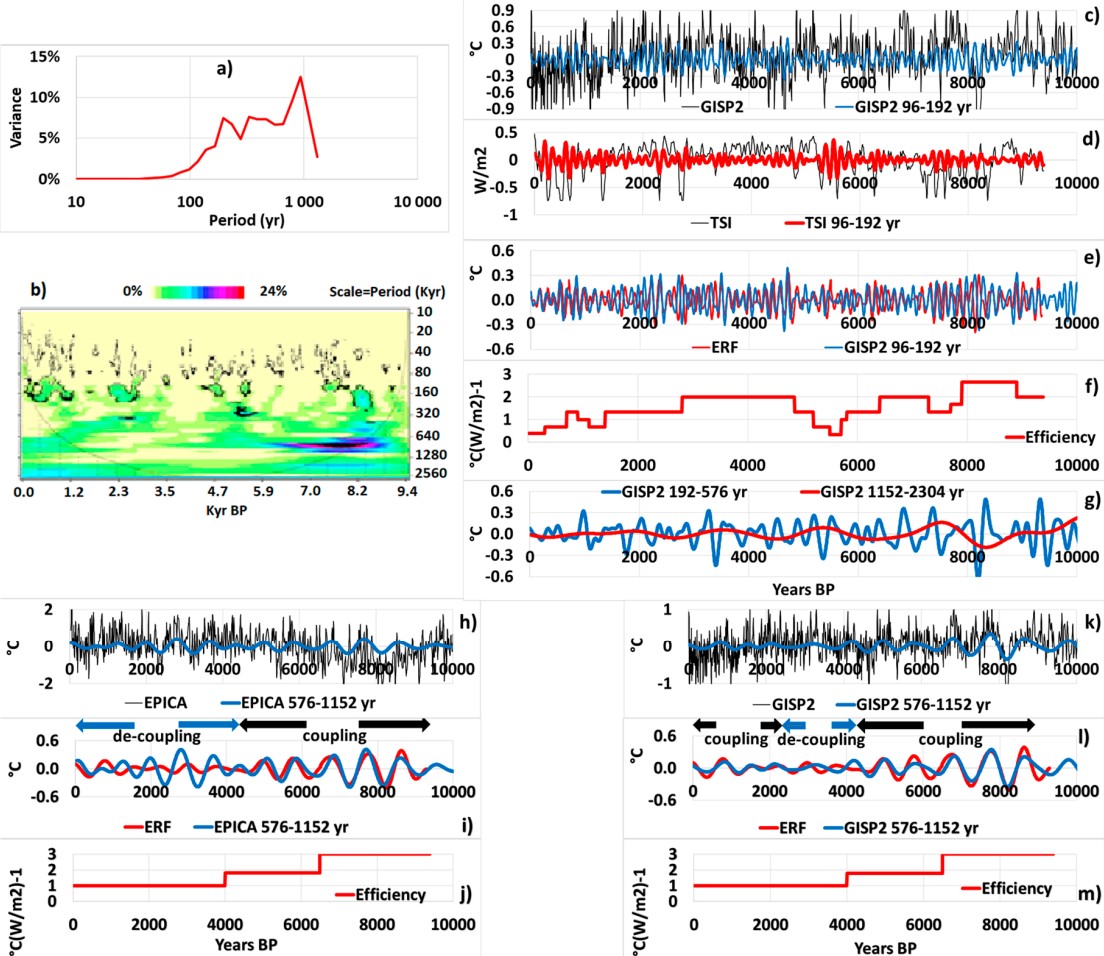

**Figure 2.** (**a**) Fourier power spectrum of the Total Solar Irradiance (TSI) during the Holocene [28], filtered below the 100 year period (the same method as in Figure 1 is used). (**b**) Cross-wavelet power spectrum of TSI and GISP2 [26,27] using the Morlet wavelet [30]. The total covariance time-averaged, scale-integrated over the whole interval of frequencies is 0.0040 °C × W/m². (**c**–**f**) Coupling of the TSI and the global temperature (GISP2) within the 96–192 yr band. A delay of 25 years is applied to forcing (**g**) the global temperature filtered into bands for which no external forcing occurs. (**h**–**j**) Coupling of the TSI with the global temperature (EPICA) [25] into the 576–1152 yr band. The signals are centered. A delay of 50 years is applied to forcing. (**k**–**m**) Same as (**h**–**j**) for GISP2.

Between 9000 and 5000 yrs BP, there is a great similarity between the GISP2 and EPICA data, the global temperature oscillating between ±0.4 °C. Between 4500 and 1500 years BP the amplitude of ERF decreases (Figure 2i,l), and the GRW disassociates from the cycle of solar irradiance, both the amplitude and the period. The GRW that is coupled to higher subharmonic modes does not weaken, which suggests the remanence of geostrophic forces throughout the gyre (internal antinodes) and along the drift currents (external antinodes). While the TSI and the global temperature are coupled again after 1500 years BP in the northern hemisphere they remain de-synchronized in the southern hemisphere. This may disclose a better forcing efficiency in the North Atlantic due to the advance of the Greenland in the Atlantic Ocean that increases the Earth latitudinal temperature gradient [8,31,32].

It must be emphasized, however, that during the last glacial maximum and the Holocene the decrease in temperature variations as the mean temperature increased while exhibiting a zonal pattern may explain some discrepancies between the hemispheres [33,34].

### 5.2.2. The 1152–2304 and 192–576 yr Bands

They are characterized by the weakness of the solar forcing so that the GRWs are strongly coupled to higher subharmonic modes. As displayed in the Figure 2g the global temperature modulation decreases regularly during the Holocene within the 1152–2304 and 192–576 yr bands, which is related to the deceleration of the polar and radial geostrophic currents.

### 5.2.3. The 96–192 yr Band

As shown in Figure 2c–f the forcing efficiency varies a lot, mainly during the periods of high solar activity during which it weakens drastically (Figure 2f). This behavior, which is opposite to what has been observed for orbital forcing (Figure 1m,n,p,q) involves the coupling strength with the higher subharmonic modes. The GRW behaves as a pure harmonic when the solar forcing is low but its amplitude increases compared to what it is during periods of high solar activity while it is resonantly forces by the Gleissberg cycle of the Sun. This involves a latitudinal elongation of the gyre because of the difference in the forcing and natural periods that results in a loss, which is only apparent, of the forcing efficiency.

## 6. Discussion

The generalization of the oceanic Rossby wave theory to long-period, multi-frequency GRWs should be a breakthrough in the understanding of the evolutions of the past climate because of the straightforwardness of the answers to some pending issues. In particular:

(1) Concerning the transition problem [35–40], the abrupt change in the dominant subharmonic mode during the Pleistocene, namely $3 \times 2^8$, when the forcing period was ruled by the obliquity results from the variation in the forcing period ruled by the eccentricity [8] (Figure 1d) while getting very close to the natural period according to the $3 \times 2^9$ subharmonic mode. The transition that occurred 0.8 million years ago supposes a 2°30′ equatorward drift of the latitude of the centroid of the gyres to adjust to the new dominant subharmonic mode.

(2) With regard to the first half of the Holocene sudden cooling always arise during the cooling phase of the 768-yr period GRW according to the $3 \times 2^2$ subharmonic mode both in the GISP2 and the EPICA data. Nevertheless, cooling arises very rapidly in the North Atlantic but more slowly in the southern hemisphere. The most striking sudden cooling event occurs 8200 yrs BP [41] as benefiting both from the advance of the polar front during the early Holocene, and a large magnitude solar irradiance into the 576–1152 yr band. Although having the most acuity in the North Atlantic, sudden cooling turns out to be ubiquitous, implying all subtropical gyres. The rapidity of both the cooling and the warming that follows involves the SST at high latitudes of the gyres where the climate is most impacted. This suggests a phenomenon of inversion of the stratification of the mixed layer, which supposes a sudden rise to the surface of cold waters with low salinity. This assumption is all the more plausible as the cooling occurs while the modulated polar current is cyclonic. So, cold waters at high latitudes of the gyres are mainly fed by the melting of the polar caps. The current resurfaces while being renewed with sea water warmer although more salty, which occurs after a few decades.

Not only is this study intended to provide new perspectives on how solar irradiance cycles influence the Earth's average temperature, probably one of the greatest mysteries of modern climate science, but also on the surface ocean circulation.

**Funding:** This research received no external funding.

**Acknowledgments:** We thank the editor and the reviewers for their helpful comments.

**Conflicts of Interest:** The author declares no conflict of interest.

**Data Availability:** The search does not use any data other than those mentioned in 'Data'.

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
