# Peer review of "Resonant Forcing of the Climate System in Subharmonic Modes"

_jmse, doi:10.3390/jmse8010060_

Round 1

Reviewer 1 Report

The author has addessed the comments of my previous review.

Author Response

Reviewer 1:

Nothing to do

Reviewer 2 Report

The response of the climate system to solar and orbital forcing is considered in terms of the resonance of the forcing with the natural periods of subharmonic modes of coupled Rossby waves. Some interesting ideas are presented here, but there are some corrections to be made and further explanation needed before the paper would be suitable for publication.  These are listed below by line number.

8 During recent decades.. raised several questions

14 waves winding around

55 parameters; 2)

56 Pleistocene; 3) .. Holocene; 4)

112 are not sufficiently constrained

119 Caldirola-Kanai (CK) oscillator

154 thermocline rise

164 Indeed, short-period QSWs

167 conditions

192 wave behave as.. other hand, shoaling of [is this what you mean, or upwelling?]

203 [similar to 192]

234 [superscript for 14C 10Be]

239 by Berger [29].

241 allows determination of

243 allows representation of the variation

245 us to represent the

276 Each band

277 corresponds to

320-321 [need to explain/justify this statement relating poleward drift to tuning]

324 1.0, up to 1.5

326-327 [need to explain/justify this statement relating equatorward drift to tuning]

336 As in the previous

348 currents are supported

415 always arise

427 after a few decades

428 intended to provide

Author Response

Reviewer 2: Corrections are made except:

Line 192: Downwelling oceanic equatorial Rossby wave is considered in https://rmets.onlinelibrary.wiley.com/doi/full/10.1002/qj.936 and in other papers when the meridional currents are converging. In the present paper gyral Rossby waves are considered but the mechanisms involved are the same when meridional currents are replaced by radial currents.

Further explanation is given:

Lines 320-321 [need to explain/justify this statement relating poleward drift to tuning] and Lines 326-327 [need to explain/justify this statement relating equatorward drift to tuning]. Further explanation is given in lines 255-257.

Reviewer 3 Report

The numerical order of the different sections needs to be modified.

The literature review needs to be updated. The article can be improved by citing the new research publications in the field, other than author's own publications.

Reference [23] seems to be the author's personal website and I doubt it qualify as a reference for the current article.

Author Response

Reviewer 3:

The numerical order of the different sections needs to be modified.

Done

The literature review needs to be updated. The article can be improved by citing the new research publications in the field, other than author's own publications.

I do not think that recent research relates to the subharmonic modes of the climate, except contrary opinion documented of the reviewer. To my knowledge, the only relevant works relate to El Nino [16, 17]. More recently, conferences has focused on “Advances in Nonlinear Geosciences”

Anastasios A. Tsonis - 2017 - ‎Science, Advances in Nonlinear Geosciences, Aegean conferences, Springer, Chap. 1. (Delay differential ENSO Model, Chekroun M., Ghil M., Neelin J.D.)

But nothing is new, Chap. 1 is referring to older papers (Tziperman et al. (1994), Jin et al., 1996). Furthermore those papers are all focused on delayed oscillators (see Lines 49-53 in the current article).

Reference [23] seems to be the author's personal website and I doubt it qualify as a reference for the current article.

OK, [23] is referring to the author's personal website that shows videos explaining the functioning of gyral Rossby waves. Obviously I could remove the reference but I think it would be a shame because the videos have primarily an educational purpose.

This manuscript is a resubmission of an earlier submission. The following is a list of the peer review reports and author responses from that submission.

Round 1

Reviewer 1 Report

This manuscript focuses on the relationship between paleoclimate temperature reconstructions and a variety of mechanisms that could be contributing to those findings, including solar forcing and net shortwave energy. The manuscript has some interesting findings, especially in the subharmonic mode response of the climate system to solar and orbital forcings, but it exhibits a number of serious issues that cause this reviewer to have strong reservations about publishing this paper.

First, the data that underlie the analysis are not properly cited. The term “climate archives” is used without reference and therefore could really mean anything. In order to have any confidence whatsoever in the manuscript, the author needs to provide a citation for these data that points to a dataset that is (1) described in a peer-reviewed publication, (2) has a large user-community, (3) has a significant number of citations and (4) has advanced the understanding of paleoclimate significantly.  

If that is the case, then the author also needs to post his analysis code of those data that was used to develop the findings in this manuscript to a publicly-available site so that the scientific community can have confidence in his findings.

Second, the manuscript asserts that Gyral Rossby Waves are the mechanism that rectifies the relationship between climate variability and variations in incoming and inferred net solar radiation of the Earth’s climate system. This assertion is problematic and distracting from the finding of subharmonic mode response of the climate system. It is problematic because the author indicates that it is an hypothesis that GRWs are responsible for these modes, but the author does not indicate anywhere how such an hypothesis could be tested. If the hypothesis cannot be falsified, it is unscientific. It is also distracting because the author does not present evidence for a GRW physical mechanism. In order for GRWs to be discussed in this paper, there needs to be a separate, peer-reviewed paper that describes these waves in detail.

Author Response

Reviewer 1:

This manuscript focuses on the relationship between paleoclimate temperature reconstructions and a variety of mechanisms that could be contributing to those findings, including solar forcing and net shortwave energy. The manuscript has some interesting findings, especially in the subharmonic mode response of the climate system to solar and orbital forcing, but it exhibits a number of serious issues that cause this reviewer to have strong reservations about publishing this paper.

First, the data that underlie the analysis are not properly cited. The term “climate archives” is used without reference and therefore could really mean anything. In order to have any confidence whatsoever in the manuscript, the author needs to provide a citation for these data that points to a dataset that is (1) described in a peer-reviewed publication, (2) has a large user-community, (3) has a significant number of citations and (4) has advanced the understanding of paleoclimate significantly.

Actually the term “climate archives” is referring to the references in the paragraph “Data”. For the avoidance of doubt each reference is recalled in the captions of the Figures, that is, where the “climate archives” are used. Those data fulfil your requirements.

If that is the case, then the author also needs to post his analysis code of those data that was used to develop the findings in this manuscript to a publicly-available site so that the scientific community can have confidence in his findings.

The software used to process the data is free to access [30]. The reference is cited in the captions of the Figures. Here again the software fulfils your requirements.

Second, the manuscript asserts that Gyral Rossby Waves are the mechanism that rectifies the relationship between climate variability and variations in incoming and inferred net solar radiation of the Earth’s climate system. This assertion is problematic and distracting from the finding of subharmonic mode response of the climate system. It is problematic because the author indicates that it is a hypothesis that GRWs are responsible for these modes, but the author does not indicate anywhere how such a hypothesis could be tested.

In fact we try to explain an issue that is pending since the works of Milankovitch (1879-1958). The concept of Gyral Rossby Waves (GRWs) that is derived from the observation of Rossby waves at high latitudes of the 5 subtropical gyres has been developed in [8, 18, 19]. Very long-period GRWs are hypothesized and their properties are formulated and demonstrated, including that they are resonantly forced by solar and orbital forcing in subharmonic modes. Everyone is free to review again these assumptions.

We advocate new hypotheses on the evolution of the past climate while challenging the role of the thermohaline circulation. These assumptions are elaborated step by step in the works cited above. This is the first time, if I'm not mistaken, that irrefutable physical evidence is being brought into the search for mechanisms responsible for climate cycles. I think that the concordance of the results is not fortuitous and cannot be a better test. Additional work is likely to be needed in the hope that this new concept opens up new avenues of research.

If the hypothesis cannot be falsified, it is unscientific. It is also distracting because the author does not present evidence for a GRW physical mechanism. In order for GRWs to be discussed in this paper, there needs to be a separate, peer-reviewed paper that describes these waves in detail.

I am sorry, peer-reviewed papers that are mentioned above describe these waves in detail. The present paper is the result of work that has been the subject of extensive development about long-period GRWs and subharmonic modes.

Reviewer 2 Report

This paper evaluates solar and orbital forcing on climate variability at long-period time scales.

The paper is scientifically sound and well explained. The methods are detailed and correctly applied. The results are well exposed and the main conclusions are in line with the presented results.

My only suggestion I that the authors expose in more detail the data used in the study, despite citing the associated papers.

Based on the above comments, this paper may be considered for publication after the minor corrections mentioned above.

Author Response

Reviewer 2:

This paper evaluates solar and orbital forcing on climate variability at long-period time scales.

The paper is scientifically sound and well explained. The methods are detailed and correctly applied. The results are well exposed and the main conclusions are in line with the presented results.

My only suggestion I that the authors expose in more detail the data used in the study, despite citing the associated papers.

OK, further information is given in the paragraph “Data”

Based on the above comments, this paper may be considered for publication after the minor corrections mentioned above.

Round 2

Reviewer 1 Report

Regarding the revision, I expect that the author should take my review seriously, address each and every point I make and substantially revise the manuscript. The "revision" that was submitted definitely did not do this. I would suggest that the author respect my review, because if not, I will strongly recommend against publishing the article.

Author Response

(The authors gave the same response as above.)
